# Developing guidelines for culturally relevant informed consent: an example from Lebanon

**Sandy Chaar**[1], **Emily Oliver**[2], **Rayane Ali**[1], **Imad Abou Khalil**[3], **Joseph Elias**[1], **Bassel Meksassi**[1], **Rozane El Masri**[1], **Thurayya Zreik**[3,4], **Michèle Kosremelli Asmar**[5], **Bayard Roberts**[4]*, **Rabih El Chammay**[6,7‡], **Felicity Brown**[1‡]

1 Research and Development Department, War Child Holland, Beirut, Lebanon, 2 BeyondText, London, United Kingdom, 3 Independent Consultant, Beirut, Lebanon, 4 Faculty of Public Health and Policy, London School of Hygiene and Tropical Medicine, London, United Kingdom, 5 Higher Institute of Public Health (ISSP), Saint Joseph University of Beirut, Beirut, Lebanon, 6 Department of Psychiatry, Faculty of Medicine, Saint Joseph University, Beirut, Lebanon, 7 National Mental Health Programme, Ministry of Public Health, Beirut, Lebanon

‡ Joint senior authors.
* bayard.roberts@lshtm.ac.uk

## Abstract

This study addresses the prominent gap in literature and practice by exploring the facilitators and barriers to informed consent and developing culturally relevant informed consent guidelines in Lebanon. Utilizing a Design Thinking (DT) framework combined with Participatory Action Research (PAR), the study aimed to: i) explore what constitutes culturally relevant informed consent in this context, according to both researchers and affected communities; and ii) use these insights to create a guideline aimed at enhancing informed consent processes for vulnerable populations involved in mental health research. The study revealed that motivations for participation, trust-building, and timing are critical yet often overlooked aspects in informed consent processes. Language and literacy barriers, along with power imbalances, present significant challenges that can be mitigated by involving community members and trained interpreters. Trust-building, especially in long-term studies, requires sustained relationships and recognizing participants' intrinsic value. Timing and clarity in consent forms, along with concise and straightforward communication, are essential for genuine informed consent. The study also highlighted the impact of gender, nationality, and community support in research participation, underscoring the need for culturally sensitive research practices. Recommendations include using audio-visual methods and the "Teach Back Method" to enhance understanding and engagement. This research emphasizes the importance of inclusive and participant-centric approaches in informed consent processes. The collaborative development of the guideline ensured diverse perspectives, leading to a comprehensive and relevant outcome. Future research should focus on testing the guideline.

**Data availability statement:** There are ethical and legal restrictions which prevent the public sharing of minimal data for this study, due to the sensitive nature of the data and the study population. Data for this study are available upon request from the Ethics Committee at the London School of Hygiene and Tropical Medicine via email (ethics@lshtm.ac.uk) for researchers who meet the criteria for access to confidential data.

**Funding:** This research was funded by UK Research and Innovation as part of UKRI Collective Fund Award UKRI GCRF Development-based approaches to protracted displacement, via grant number ES/T00424X/1 to all co-authors (REL, FLB, JE, BM, RA, SC, MM, MKA, BR, REC, NSS). The funders had no role in study design, data collection and analysis, decision to publish, or preparation of the manuscript.

**Competing interests:** The authors have declared that no competing interests exist.

## Introduction

In recent years, the discourse surrounding research methodologies has expanded to include a critical examination of how research can unintentionally support unfair practices, especially concerning vulnerable populations. This exploration aligns with contemporary discussions on decolonizing research [1,2]. The primary goal of decolonizing research is to include and value the perspectives and knowledge of indigenous and marginalized communities [3]. This involves rethinking and transforming research methodologies, practices, and frameworks to challenge the dominance of Western, colonial paradigms. One central concern highlighted in this discourse is the notion of research being extractive. This refers to a paradigm where researchers collect data from participants to address their own inquiries, utilizing methods that align with their worldview, and enjoying the benefits in the form of publications and career advancement [4]. Often, this process occurs without meaningful consultation and active participation of the affected communities, resulting in a one-sided approach that may not necessarily translate benefits back to the participants.

This concern aligns with the call for a more inclusive approach and participatory model in scientific inquiry, ensuring that the voices of affected communities are actively incorporated into the research design and implementation [5].

One area in which methods to decolonize research could be first explored is the process of informed consent. Consent is a dynamic process involving researchers approaching potential participants, providing information, and addressing questions to ensure understanding. Participants then make an informed decision about participation by signing a consent form [6]. The importance of informed consent in research participation cannot be overstated, as it highlights the value of respecting individuals' autonomy [7]. International regulations, such as the Declaration of Helsinki [8] and the Principles of Biomedical Ethics [9], emphasize the importance of ensuring that potential participants comprehend the information before giving consent [10].

In research in low- and middle-income countries (LMICs), there is growing concern about how well individuals understand informed consent, particularly due to the vulnerability of participants and the potential for research exploitation [11]. It has been emphasized the importance of minimizing the exploitation of participants enrolled in the research by providing fair material benefits, yet this is controversial as excessive offers may distort decision-making and lead individuals to participate against their better judgments [12]. In the context of humanitarian settings, research finds that the use of written consent may not be universally effective in ensuring genuine engagement and understanding, particularly in societies where oral discussions are customary for important decisions [13]. The use of multimedia resources, such as videos or websites has been limited to date, but it has been argued that their potential benefits in improving understanding warrant further exploration [14–19]. Additionally, with the increasing number of studies in humanitarian contexts among very vulnerable populations living in complex situations, comes significant concern about the direct benefits of research for affected populations, and the potential harm among participants if improvements are not achieved. Hence, the Inter-Agency Standing Committee [20] stresses on the importance of adequate informed consent in mental health and psychosocial support (MHPSS) research and of ensuring that the study provides direct benefit to affected people in emergencies.

Research participants and researchers report different experiences with informed consent. Some participants find it difficult to grasp the purpose, methods, and potential risks and benefits of a study. In addition, there are mixed participant perceptions on the advantages and disadvantages of lengthy and detailed Participant Information Leaflets/Informed Consent Forms (PIL/ICF) [21]. Researchers on the other hand realize the importance of detailed information but cite it as typically lengthy and complex, which creates a significant time burden [11].

Moreover, some participants may not be aware of their right to withdraw from the study at any time, and deficiencies in the consent process may be due to the information providers rather than the participants [15]. Some ethical concerns surrounding research consent also focus on the power dynamics between researchers and participants. The concept of "reciprocal dialogue" was introduced as an ethical methodology to tackle these challenges, which emphasizes the importance of mutual trust and equality between researchers and participants [22].

Very little research has been conducted regarding how to improve the process of informed consent, especially when working with vulnerable populations such as refugees and those exposed to humanitarian emergencies, to address problems related to human interactions [23]. Design Thinking (DT) processes hold promise in developing more sustainable and effective solutions to public health research challenges [24]. This is achieved by placing a strong emphasis on comprehending user needs and perspectives, and dynamics in the relationship between researcher and participant. Here, we suggest DT can be enhanced by integrating Participatory Action Research (PAR) which compliments DT's action-orientation while strengthening the focus on participation and reflexivity [25]. PAR is a research approach that actively engages the individuals impacted by the research to address real-life problems through collaboration and drive social change [26]. It emphasizes the collaboration between researchers and participants, allowing them to collectively recognize issues, develop solutions, and implement actions for enhancement.

This study addresses this prominent gap in literature and practice by using a DT framework combined with PAR methodology to frame the problems and potential solutions to develop culturally relevant informed consent guidelines collaboratively between researchers and research participants. The objectives of this study were to: i) explore what constitutes culturally relevant informed consent in this context, according to both researchers and affected communities; and ii) use these insights to create a guideline aimed at enhancing informed consent processes for vulnerable populations involved in mental health research.

## Methodology

### Setting and partnership

Lebanon currently grapples with one of the most significant refugee crises globally, hosting around 1.5 million displaced Syrian, one of the largest refugee populations per capita globally. As a result, a quarter of the population in Lebanon are refugees, many of whom are reliant on humanitarian assistance [27,28]. This crisis is exacerbated by a severe economic crisis in Lebanon, forcing 44% of the total Lebanese population below the poverty line, and which has been described as one of the most severe financial crises globally since the mid-nineteenth century [27,29].

In response to these challenges, various international and local actors are actively involved in providing support, particularly focusing on health, education, social services, and water supply. There are also research initiatives aimed at identifying and addressing the needs of host and displaced populations [30]. However, refugees have expressed frustration over the volume, perceived benefit and potential exploitation of participating in the research [31,32].

This PAR study involves DT processes as a collaboration between an international NGO heavily involved in research in humanitarian contexts, a storytelling & co-production not-for-profit organization, a DT consultant, a UK-based university, and a Community Advisory Board (CAB) in Lebanon. The CAB were previously part of another research project involving a psychosocial intervention. They were contacted by the focal points in the community who already had direct contact with them.

To ensure the rigor and transparency of our qualitative study, we adhered as much as possible to the COREQ (Consolidated Criteria for Reporting Qualitative Research) checklist, which guides the reporting of key components of the study such as the research team, data collection methods, and analysis procedures. The completed COREQ checklist is included in the S1 Checklist, ensuring all relevant components of the study are transparently reported.

This work was part of the GOAL research project which was a collaboration between War Child Holland (Research and Development Department and War Child Lebanon), Beyond-Text, the National Mental Health Program of Lebanon, ABAAD, St Joseph's University of Beirut, and the London School of Hygiene and Tropical Medicine.

## Design

We conducted qualitative semi-structured interviews and focus group discussions combined with DT workshops with researchers and CAB members in Lebanon. Employing ELRHA's Participation in Humanitarian Innovation Toolkit and other DT exercises served to construct personas for both researchers and participants, dissect challenges encountered at various stages of the research process and at various interpersonal and intrapersonal levels, and formulate solutions aimed at enhancing the overall participant experience while ensuring ethical standards for research are upheld.

## Sampling and Study Participants

A total of 22 participants took part in the study.

This included 11 CAB members: 10 females and 1 male, of Palestinian (n = 3), Syrian (n = 4), and Lebanese (n = 4) nationality, and coming from two geographical regions of Lebanon-North and Beqaa governorates. They were invited through their existing participation in the CAB developed for a prior large-scale research project designing and evaluating a psychological intervention for vulnerable families.

The sample also included 11 researchers from the International NGO War Child Alliance, eight females and three males from Lebanon (n = 6), UK (n = 1), Uganda (n = 2), and the Netherlands (n = 2). These participants were invited to take part via emailing all researchers within the global research team.

In order to determine the desired level of participation of the CAB members and NGO staff in this study, ELRHA's Participation In Humanitarian Innovation Toolkit was used. This tool is designed to offer the necessary expertise for creating humanitarian innovation journeys that prioritize the involvement of individuals affected by crisis [33]. Based on this resource, the desired level and type of participation was evaluated and determined for every individual by the DT consultant. The participation matrix suggested by ELRHA, proposes three types of participation: consultation, partnership and ownership [33]. Each type encloses varying degrees of engagement to distinguish how much decision-making power an individual has over the project. While applying this approach, we observed that researchers from the core team (SC, EO, IAK) encountered no obstacles in assuming leadership roles in the study. Additionally, JE, BM, and RA took on co-creation partnership roles. Conversely, while CAB members were keen on taking an ownership type of participation, hindrances such as limited resources (time, technology, etc.), technical research expertise, and language proficiency led to a partnership level of involvement described as collaboration. Hence, they participated in information gathering but not in planning, analysis, or synthesis activities.

Inclusivity in global research: Additional information regarding the ethical, cultural, and scientific considerations specific to inclusivity in global research is included in the S3 Checklist.

## Procedures

To enable effective analysis of the framing of problems and potential collaborative solutions to IC collaboratively by research participants and researchers, DT was used as a framework that enables participation in co-design by people outside of the design profession, including for innovations related to health [34]. DT involves a human-centered approach to problem-solving, typically encompassing (1) empathizing, (2) defining the problem, (3) ideating, (4) prototyping, and (5) testing solutions iteratively. For the sake of this study, we planned to complete steps 1–4. Testing was not part of the scope of this study and is planned as next steps. Table 1 below outlines the steps and methods used in this study.

### Step 1-Literature review

In developing the methodology for this study, we used insights from a non-systematic literature review on facilitators and barriers to informed consent. The review encompassed a broad search focusing on prior studies addressing consent issues within similar research contexts. Recurrent themes identified in the studies helped to inform the formulation of Key Informant Interviews (KIIs) and Focus Group Discussions (FGDs). This approach ensured that the study design and data collection methods were grounded in the perspectives and concerns highlighted in the existing literature.

### Step 2-DT exercises with core team

DT exercises and reflections were done with the core research team based on "journey mapping" and "AEIOU" exercises – A (activity), E (environment), I (individual), O (objects), and U (understanding) [35]. These exercises were adapted and facilitated separately and adapted to respondent type to the research participants and researchers. First, the journey mapping exercise was used as a method for understanding and enhancing the informed consent process. It is a qualitative research technique that visually illustrates the journey of participants and researchers through the informed consent process, capturing their experiences, emotions, and interactions at each stage, with the aim of identifying ways to enhance the process for

Table 1.  Research steps overview.

| Steps | Who | What |
|---|---|---|
| **1.Literature review** | *SC* | Developing the KII & FGD guides |
| **2.Design thinking workshops** | Core Team (SC, IAK, JE, BM, RA, EO) | Journey Mapping & AEIOU. Fed into FGD guides for next steps |
| **3.Focus Group Discussions (FGDs)** | Core Team (SC, IAK, JE, BM, RA, EO), NGO researchers, CAB | FGD guide informed by DT findings |
| **4.Key Informant Interviews (KIIs)** | CAB only | Follow up from FGDs to give more space to share personal experiences on themes arising from focus groups |
| **5.Data analysis** | Core Team (SC, BM, JE, RA, EO, IAK) | • Thematic qualitative analysis of Steps 2,3,4<br>• Identifying themes<br>• Developing a Problem Statement |
| **6.Feedback sessions** | CAB and Researchers | Presented findings from both groups of participants for validation & elaboration on potential solutions |
| **7.Ideation session** | Core Team | Reflected on results of analysis as well as the feedback sessions, and how this would inform the guideline and developed a matrix, Developed a Guideline |
| **8.Prototyping** | Core Team | Full guideline drafted based on the feedback of the Researchers and CAB |

users and promote participant understanding, autonomy, and engagement [36,37]. Next an AEIOU exercise was conducted which breaks down a problem into different aspects. This framework is used to better understand and generate creative solutions to the problem to solve real-world problems by looking at these different aspects. The insights we gathered from the design thinking exercises were integrated into the focus group discussions (FGDs) and interview guides. The research team piloted and adjusted the guides in various workshops to ensure that the questions were clearly and accurately translated between English and Arabic, while retaining their intended meanings and contexts

### Step 3 and 4-FGDs and KIIs

FGDs and KIIs were then conducted, focusing on the participants' experiences of consenting to research and perceived benefits and barriers to participating in developing alternative solutions to improve current consenting processes. FGDs were conducted separately with NGO researchers and the CAB, lasting about 1.5 hours with both groups (see Table 2 for participant characteristics). These FGDs also incorporated the DT exercises conducted in Step 2. All DT exercises were facilitated by a DT expert, all KIIs were carried out by SC, JE, RA, BM from an international NGO. Each interview lasted for approximately 30 minutes and involved private conversations with the CAB members. KIIs with CAB members were conducted after the CAB FGDs to expand upon the insights gained from the group discussions. This approach allowed the CAB members to express their personal thoughts and experiences more freely, without the potential influence of group dynamics. All FGDs and KIIs were audio recorded with participant consent.

### Step 5-Analysis

Audios of all interviews, FGDs and DT workshops were translated and transcribed to English by bilingual research team members. To guarantee accuracy, transcriptions underwent verification by additional members of the core research team, and a ten percent sample of transcriptions was subjected to back-translation. Multiple researchers analyzed interviews and FGDs collaboratively using the blind coding feature provided by Dedoose, in line with the project's commitment to collaborative coding [38]. Transcripts were coded by pairs of researchers (SC, IAK, EO, JE, RA, BM) based on a codebook developed from an inductive approach, while focusing on the research questions. The team regularly met to discuss the codes and themes. An analysis workshop involving all members of the Core Research Team was conducted to identify key themes and recommendations from the data.

Table 2. Participant characteristics in FGDs and KIIs.

| Group | Total participants | Gender | Nationality |
|---|---|---|---|
| FGDS - researchers | 11 | 8 females 3 males | Lebanese (6) Ugandan (2) Netherlands (2) United Kingdom (1) |
| FGDS - CAB KII - CAB | 11 | 10 females 1 male | Lebanese (4) Syrian (4) Palestinian (3) |

### Step 6-Feedback sessions

After the analysis of the data based on key informant interviews, FGDs and DT workshops, feedback sessions were conducted in July 2023 to clarify the problem further and reflect on further solutions. Tentative findings and recommendations from the data analysis were shared with all participants (NGO researchers and CAB members) during validation workshops and refined as needed based on feedback.

### Step 7-Prototyping

To identify and ideate the prototype of culturally relevant IC, the core research team reflected on the results of the analysis to inform the development of a guideline for improving IC processes. This guideline is to be tested.

### Ethics

Ethical approval was obtained from Saint Joseph University in Beirut (ref: USJ-2020-224, 19/01/2021), and the London School of Hygiene and Tropical Medicine in London (ref: 22766, 13/01/2021). Prior to conducting interviews and focus groups, participants were provided with a Participant Information Sheet that outlined the study's purpose and scope. Written consent was obtained from every participant. To safeguard their identity, all identifying details in the transcripts were anonymized, and numerical codes were assigned to each transcript.

## Qualitative research findings

We identified five different themes from the data stemming from the DT workshops, FGDs and KIIs. These are summarized in Table 3 and described in detail below. We have also visualized their inter-connectedness in |Fig 1.

Table 3. Key themes and sub-themes from the qualitative research.

| Theme | Sub-theme |
| --- | --- |
| Motivations and incentives | Factors influencing participation |
| | Incentives and potential coercion |
| | Barriers to participation |
| Trust and rapport | Trust |
| | The approach of the researcher |
| | Trusted actors, channels and locations |
| Informed consent delivery | Time constraints |
| | Clarity of information sheet |
| | Right to withdraw |
| | Dialogical process |
| Gender, nationality, community support and power dynamics | Gender |
| | Nationality, stigmatization and exclusion |
| | Community and family support |
| | Unequal power dynamics |
| Additional recommendations | Participant involvement |
| | Clarity in materials and accessibility |

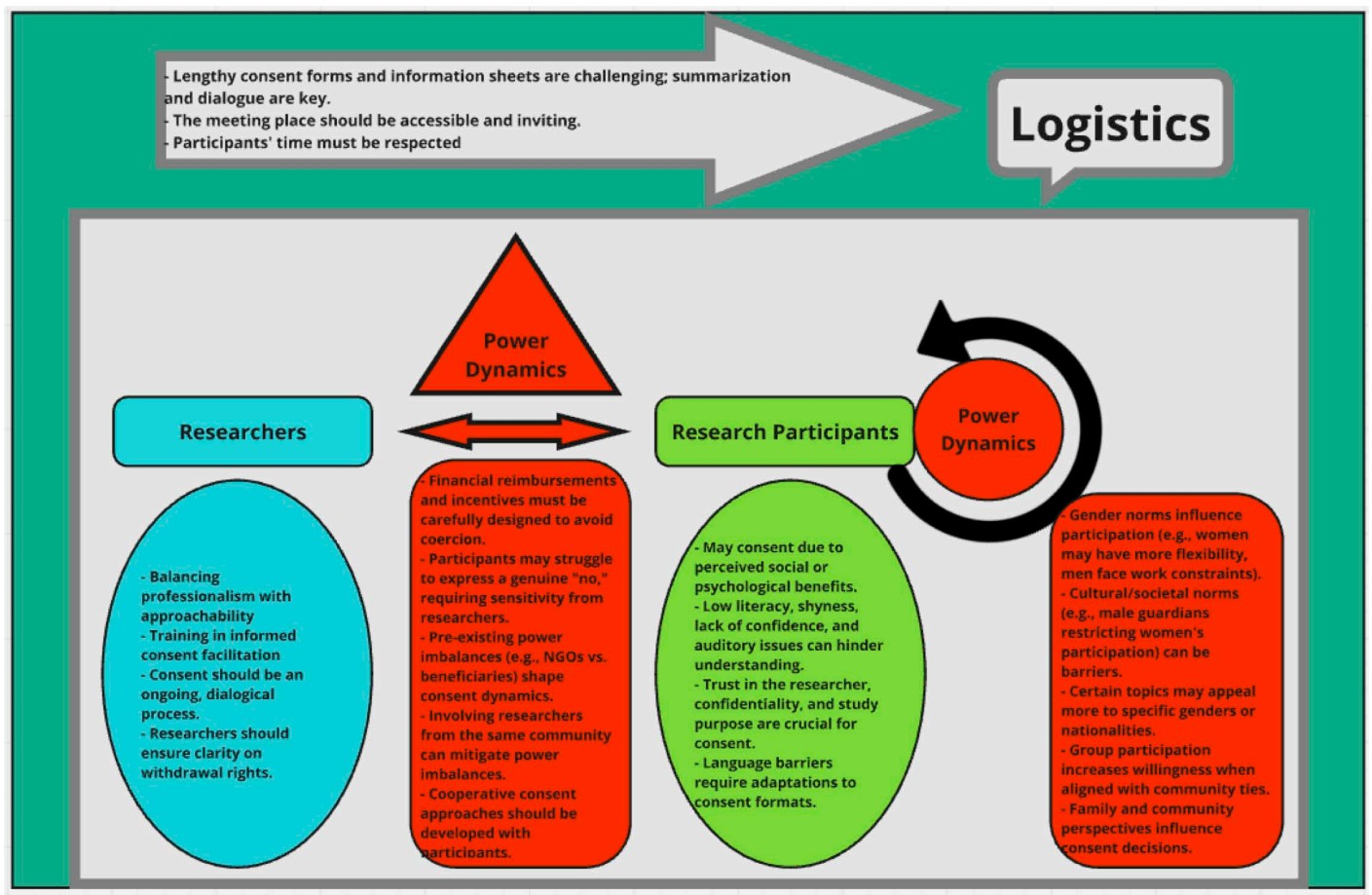

## Relevant Informed Consent

*This framework outlines key factors influencing the informed consent process in research, categorized into five dimensions: (1) **Researchers**, focusing on their approach, training, and role in facilitating consent; (2) **Power Dynamics Between Researchers and Participants**, addressing imbalances, financial considerations, and community trust; (3) **Research Participants**, highlighting barriers such as literacy, confidence, and perceived benefits; (4) **Power Dynamics Between Participants Themselves,** considering gender, cultural norms, and group dynamics; and **(5) Logistical Factors**, emphasizing the place, time and length of the informed consent.*

**Fig 1. Conceptual framework for contextualizing culturally.**

## Theme 1: Motivations and incentives for research participation

**Factors influencing research participation.** CAB members reported that people are more likely to consent to a research study if the topic of the research benefits them or their families, such as improving their education, career, or health. The idea of helping others through the research also motivated them to participate. Parents were more likely to agree to participate if their children were also involved in the study and showed an interest in it. Even when there is no financial benefit, CAB members were willing to participate if there is a perceived psychological or social benefit for them, including an opportunity to express ideas and feelings, or learn something new, travel to a new place, have a change of environment, or meeting new people: "*We get to know new concepts. We get to meet new people. We get mental health support. We go back and talk with our relatives and neighbors about what is happening here*" *(CAB Member).* Additionally, they expressed additional motivations for consenting such as the satisfaction of contributing to the community, making an impact, and helping others.

**Incentives and potential coercion in participation.** Nonetheless, financial reimbursements were reported as crucial for some CAB members, especially in difficult financial situations. In the feedback sessions, some emphasized that monetary compensation was not always the most important thing: "*...I'm here to learn more, and to express myself. Money isn't important.*" *(CAB member).* However, another member expressed that: "*In the current economic situation, motivation would be less if there is no financial incentive.*" Inversely, some NGO researchers pinpointed disadvantages to providing monetary compensation to participants since they: "*... would just agree, because they are desperate to receive anything in their state of mind.*" *(NGO Researcher).* Consequently, they stressed that participants may ignore their rights to get some perceived benefits and may feel pressure to participate to maintain financial benefits. Overall, NGO researchers highlighted that incentives do not need to be monetary and may consist of essential contributions to address barriers to participation such as transportation fees, mobile phone data, or childcare. As one researcher suggested, "*compensation is important, it doesn't have to be material all the time - it's simply what the participants need to be able to participate.*" Providing such compensations was seen as helpful in making participation more accessible and inclusive; however, it was essential to ensure that individuals are not forced or influenced solely by incentives.

"Yeah, really this challenging balance of not forcing them by that incentive, but really incentivizing them to participate [...] Yeah, is the incentive, just so strong that if you're so vulnerable, that actually anything that you get is better than nothing. And then you're willing to give away your right to not participate just due to the lack of resources that you have? [...] And also, even if you're told, like, if you withdraw, or if you don't join, it will not have any consequences on annual receiving nor any other services exactly as P5 said, often that perception is still there, like, oh, but maybe it will have some impact somewhere and the organizations talk among each other and they might tell each other" (NGO Researcher)

In addition, researchers acknowledged that participants may think that they will be rejected from upcoming benefits if they declined to participate in the current activity, and, as with financial incentives, this can cause undue pressure to take part if not managed well.

"*If I say no, is it going to impact whether I can attend, whether my child can attend, or whether they're going to take their funding away? Are they going to not have activities anymore? And so there's pressure to participate in order to maintain services or maintain the interventions that are being done. So I think that can be an obstacle to people saying no as well.*" *(NGO Researcher)*

**Barriers to participation: competing priorities and communication challenges.** CAB members may sometimes face competing priorities in their personal life, causing them to prioritize other things over their participation in research. For example, one participant stated that they had to prioritize something else when their schedule became tight: "*… but after a while, a new thing popped up and my schedule became tight. I took what I wanted from my participation, so I prioritized something else.*" (CAB Member).

Additionally, low literacy and fear of not understanding the research topic may have led CAB members to a lack of motivation to participate. Shyness, lack of confidence, and auditory problems were suggested to hinder participation. With regards to literacy, CAB members expressed that they might be shy to participate, especially if they feel that they might not understand the information presented to them: "*…The language as well. Sometimes, some researchers might mumble some words that we don't understand, and some people may feel shy to ask about what they mean.*" (CAB member). Researchers acknowledged: "*…the challenges that we face is how to translate or convey what we want to do in such a way that is completely understandable*". (NGO Researcher). Findings from the researchers also suggested that in research studies, there can be a potential loss of information and meaning in translation, especially if the participant's dialect or first language differs from that of the researchers, however having a confident and supportive facilitator can help them overcome these barriers. In the feedback sessions, CAB members described there being no problem if they can ask the researcher for clarification if they don't understand something.

## Theme 2: The crucial role of trust & rapport building in research participation

**Trust.** The recognition of trust as a crucial element was paramount. CAB members mentioned several factors that influence trust such as being clear about the purpose of the research. When they understood who was gathering their information and why, they felt more comfortable and confident in providing informed consent. Clarity and understanding of all aspects of the research process helped in consenting, as mentioned by one CAB member: "*…I do not buy fish in the sea*" (a metaphorical way of expressing skepticism when transparency is lacking). They appreciated when all conditions are written, explained, and anonymity is provided, as "*…these two factors made us agree to participate in the research: honesty & safety*" (CAB Member).

**Researcher's approach.** CAB members emphasized the significance of researchers being welcoming and warm, as this created a positive atmosphere and built trust with participants: "*…I don't like to feel that I am beneath people. That some individual came to give me information. They should use dialogue with us, so I would feel like a participant, not just a recipient. Some people would just state information without listening to your opinions, so you would feel unheard, and you wouldn't be interested in attending the next session.*" (Feedback Session, CAB Member). They also reported that a fostered sense of comfort is crucial, allowing them to relax and engage more openly in the consent process, and that researchers can create this by being professional but also approachable as humans, such as by taking time to build human connections, allowing participants to '*breathe a bit*' beyond just focusing on the research activities, and giving space for humour and telling jokes. This was highlighted in the feedback sessions by a CAB member:

"*You give us a chance to speak, so this means you respect me and respect my time. Also, you're answering all my questions in a respectful way, even if my questions were insignificant. You're giving me time to speak. And an important thing is that you told us we have the complete freedom to share or not.*"

However, some NGO researchers stressed that rapport building should not be obtained before taking consent, as it might affect the consent process. *"…So, it might affect because the person is getting on a level that "ohhh this person is nice"* as one NGO researcher indicated. Another one said: *"[...] I think that a lot of us, like us as researchers, take the stuff way too seriously. And we take ourselves - not the ethics of consent- but we take ourselves way too seriously. The whole thing about professionalism. I think there's a degree of being personable, but that you can do it without compromising either the data collection or the research." (NGO Researcher).* In the feedback sessions, it was emphasized that the ideal researcher is expected to strike a balance between being personable and maintaining professional standards during the research process: *"it is a delicate balance to find, but I think it's important to make sure that people are able to say no. But also that people are able to ask questions." (NGO Researcher)*

**Trusted actors, channels and locations.** The CAB member's personal trust in the researcher was mentioned to encompass trust in their confidentiality, trust in the overall process of their work, credibility of the conducting organization, and trust in both the consent procedure and the purpose of the study. Also, the inclusion of local trusted actors in the outreach efforts was helpful. CAB members were more likely to trust research conducted through familiar and trusted channels, enhancing the credibility and reliability of the research. The meeting place also plays a crucial role as it needs to be attractive and accessible, creating confidence and making the cost of participation affordable: *"…And we know where you came from, and we know the location of the session. If the session was conducted in somebody's house, I probably wouldn't attend." (Feedback Sessions, CAB Member).* Therefore, researchers were asked to carefully consider the location of the meeting and ensure that it is a place that the participants feel comfortable with and can easily access.

According to CAB members, lack of trust poses a significant barrier to participation, extending beyond informed consent. Surprises during research activities erode trust and diminish willingness to engage. One participant expressed: *"…Sometimes we participate in sessions and things get delayed, we get surprised each time with new facts, but today I really liked how clear you were with us. However, such experiences make your trust less. (CAB Member)."* Past research experiences influenced trust in new studies, with unsatisfactory encounters breeding caution and skepticism.

## Theme 3: Informed consent delivery: Importance of time, training and contextualization

**Time constraints.** Time was identified as a significant barrier to obtaining informed consent in a research project. NGO Researchers in this study reported that researchers sometimes seem to assume that time will not be perceived as a burden for participants, however this was described as incorrect *"…we take it for granted… Oh, yeah, they're just sitting in a refugee camp. Of course, they'll want to participate in our project, but actually it does place a burden' (NGO Researcher)".*

One researcher shared their experience with a research participant perceiving the detailed explanations and time-consuming nature of the provided informed consent tool as a *"marriage arrangement"*. This comparison reflected their perception of the consent process as excessively intricate and unnecessarily burdensome. Consequently, they went on to ask: *"… have you come to ask for my hand in marriage?" (NGO Researcher).* Hence, the researcher highlighted the importance of informing the participants of the length of the informed consent process and how much time is needed from them to go through it and why. It was stressed on during the feedback sessions that it is very important to be transparent about time commitments needed, and respect participants' time: *"…So I think, yeah, when we talk*

*about the power dynamics, I think we should be very respectful of people who said they want to participate in what we ask them to do.*"*(NGO Researcher).* Researchers suggested that sufficient time should be allocated for the informed consent process, allowing for introductions, question-and-answer sessions, and discussions. They believed that rushing the process is counterproductive and recommended allocating sufficient time for participants to become familiar with the research, seek advice from others, ask questions, and reconsider their decision. Conversely, some researchers expressed frustration and described the consent process as "annoying" and "lengthy" in time; hence, leading the researchers to rush through the process, treating it as a mere administrative task. As a result, participants might not be sometimes not fully informed about the purpose and implications of their participation, reducing the quality of their consent. *"… But just almost like a tick the box thing and almost like are you okay with participating? Everyone is fine? Yes, great. Let's go. Because it is very annoying, and it takes a lot of time" (NGO Researcher)"*

**Clarity of information sheets.** The information sheet was described by both researchers and CAB members as important to outline advantages and disadvantages of the research and have contact information for clarification. Overall, CAB members held in high regard a clear and honest explanation of the informed consent, which enables them to make an informed decision and feel protected by the consent they signed. They also emphasized that the information sheet: "*...should not be that complicated, just simple and to the point.*" *(CAB member).* Researchers reported that lengthy information sheets are challenging due to the potential for participants to become bored or fail to understand them. They suggested the use of summarization and engaging in conversations to supplement the written information sheet. Therefore, researchers emphasized the importance of prioritizing participants' understanding and obtaining their consent for ethical research and also stressed the need to inform participants about the use of their personal information and the sharing of research results.

**Right to withdraw.** CAB members reported that they understood their right to withdraw from the research at any time without guilt or the need to provide a reason. The concept of feeling '*pressured*' is contrasted with feeling '*comfortable*' to participate, highlighting the importance of creating an environment where participants have "*the autonomy to participate.*" *(CAB Member)* and make their own decisions. NGO researchers, however, reported difficulty in discerning a genuine "no" for participation, from misunderstandings, and knowing how to navigate this while ensuring they prioritize individuals' autonomy. During the feedback sessions, researchers clarified that participants are not obligated to provide a reason for their withdrawal from the research, but it can be informative to know about their reasons as part of the learning process: "*.... So it would be great to ask people, even though making sure that they know that there's no necessity to provide a reason, it would be interesting to understand why people drop out. Because there could be something, say something about the intervention, they could say something about the instruments that we use, it could say something about the process that we apply* (Feedback sessions, NGO Researcher).

**Dialogical process.** NGO Researchers highlighted that it is crucial for researchers to understand the importance of informed consent as a dialogical process involving conversations with participants, rather than a mere checklist. Thus, training on the delivery and facilitation of informed consent is a must. One researcher suggested that: "*...actually the whole process and the training on what to do with this form, how to convey it, to make it clear* [should be done]*".*

While researchers highlighted that informed consent is an ongoing process throughout the study, some CAB members viewed acceptance as a commitment and believed that once consent is given, it cannot be withdrawn. They drew parallels between job contracts and informed consent, highlighting the notion that once consent is given, it cannot be withdrawn, similar to

a commitment made in a job contract. For example, one CAB member stated: *"…it's like a job contract you do for a while, and you cannot go back if you consent."* This perspective suggested that participants could perceive informed consent as a binding agreement, comparable to the obligations in a job contract. In the feedback sessions, one CAB member acknowledged that if their expectations were unmet over time, they would withdraw but emphasized that as long as there is a signed agreement, they feel obligated to adhere to it: *"…Then I would withdraw, since there is no contract. As long as there is a signature, I'm bound by it."*

### Theme 4: Gender, nationality, community support and power dynamics in research participation

**Gender.** According to the participants, gender plays an important role in research participation, as some people may be more comfortable sharing personal information with individuals of the same gender. Women are perceived to have more domestic responsibilities than men and are often able to work around them to participate in research, whereas men may be unable to participate due to work responsibilities. However, there may be cultural or societal barriers that prevent women from participating, such as men not allowing their wives to participate in research or culturally conservative views towards mixed-gender settings. Ultimately, the topic of the research itself may also influence gender differences in participation, as certain genders may be more familiar or interested in certain topics.

**Nationality, stigmatization and exclusion.** Similarly, nationality could be influential as the topic of research may be more interesting or relevant to some nationalities, leading to varying levels of motivation to participate. In addition, participants reported that feelings of exclusion or discrimination based on nationality, race, or refugee status can be a significant barrier to obtaining informed consent. If individuals perceive that they are being stigmatized or marginalized, they will be more reluctant to agree or consent to participate in the research. While this may not directly relate to the consent tool itself, it can significantly affect individuals' overall decision-making process as one participant pointed out: *"… Now we are here all participating together, but some people are racist they would not accept different nationality especially in a group of a certain majority, if a person does not feel welcomed, they will quit, this has an effect, I guess." (CAB member)*

**Community and family support.** The perspectives of family members and other community members, such as neighbors, was described by CAB members as a reason to influence consent decisions at times, but this is not always the case. It was reported that some CAB members are more willing to participate in research or share knowledge when it contributes to the growth and development of their community, and that consenting to research is more likely to happen when they know that they will be participating with others from their same community. In these cases, community support was also seen to play a role in addressing literacy challenges by reading and/or explaining information to each other.

**Unequal power dynamics.** According to researchers, the pre-existing unequal power dynamics already present between NGOs and beneficiaries due to the structure and nature of humanitarian aid and services was felt to play out during research projects involving vulnerable populations. They highlighted how repeated research on vulnerable populations can perpetuate unequal power dynamics, where researchers and NGOs dominate the narrative and resources, potentially overshadowing the voices and agency of the displaced populations themselves, leading to vulnerable participants not believing that their voice matters: *".. we are not completely genuine within the entire sector, in giving people their voice or ensuring people have their own voice." (NGO Researcher).*

In the feedback sessions with CAB members, the impact of these power imbalances was noted, such as when participants perceive higher authority or status in the data collector or when gender dynamics influence responses. However, they highlighted how important it is for them to share ideas and knowledge with the researcher knowing that it will benefit the study. They believed that even with their humble knowledge, they can contribute to developing important things.

CAB members and NGO researchers talked about the fact that research participants and researchers come from different social norms and backgrounds. Hence, their perceptions of things might be different, and this should be taken into consideration equally.

Participants expressed several ways to mitigate these power dynamics. Researchers suggested that matching the data collector's background or level with the participant's status can help mitigate bias and ensure a better quality of data. One researcher stated that "*... if it's a group for men, then you would want the research team to be logically maybe a man, maybe for balance, you can just put a woman, but it would be good, you know, to have a male figure in a group discussion of men. If it is a group discussion for women, you think the women will respond more to a fellow woman... And then maybe if it's a high-level data collection, where you have maybe high-level high risk reputable people in the community, then you would equally take a research assistant, who is also at that level, who may not be looked at as somebody who is not worthy to collect the data from that person.*"

The presence of community members who act as "counselors" might improve the CAB members' comfort and understanding of informed consent. Additionally, participants suggested that facilitating the informed consent process in a group format can be helpful when people start asking questions which can be supportive to others in the group who are shy to voice their thoughts out, however they acknowledged the possibility of people consenting due to wanting to please others when consenting in a group format.

## Theme 5: Additional recommendations

**Participant involvement.** Researchers stressed that addressing this issue involved putting participants at the center of the decision-making process and to make informed consent something that is '*driven by the people*' *(NGO Researcher)*. They suggested a cooperative informed consent approach in which potential research participants and researchers draft contextually relevant consent processes and documents together and ensure that they are adapted to the appropriate context, social norms and literacy. To ensure the clarity of materials, it was proposed "*...to involve a third party, such as a layperson, as a reference point to check if the information sheets are understandable.*" *(Feedback Sessions, NGO Researcher)*. Involving researchers from the same community was noted as an effective way to address many of the challenges outlined, along with raising awareness and knowledge of the research and its potential societal benefit and shifting decision-making power to the people in the context. Ultimately, researchers proposed that when participants feel part of the research and see potential personal benefits, they are more likely to perceive the value of the project and consent to participate. In addition, the informed consent process should be mutually beneficial and prioritize transparently outlining contextually relevant benefits and risks, with ongoing feedback to ensure a balanced presentation of expectations and benefits. Researchers emphasized the importance of showing evidence and practical information about what will happen and making ongoing consent and the right to withdraw clearer for more voluntary consent.

Suggestions were made to include multiple stages of consent, involving both group information sessions, and individual conversations, with the optimal structure depending on the

methodology and recruitment process. However, it was emphasized that in all cases the final decision to participate or decline should be made on an individual basis to ensure autonomy: "…*So I am thinking the group information sharing can apply if the unit of research you're thinking of to recruit is an organized group. So there you can give information generally in a group, but still when it comes to consenting, the assumption is everybody will still have the autonomy to decide on whether to participate or not to participate.*"

**Clarity in materials and accessibility.** In terms of more accessible formats, participants recommended exploring alternatives to paper-based consent, such as oral delivery, voice notes, visual aids, and a combination of pictures/videos and explanations, in order to enhance comprehension and engagement in the informed consent process. However, participant preferences varied widely, and they still stressed on the utility of also having pen and paper methods.

Researchers stressed on the fact that formats of informed consent should be tailored to suit the population, making it as meaningful and inclusive as possible. Similarly, multiple communication options for follow-up were seen as important to confirm the understanding of the informed consent. During the feedback sessions, researchers suggested participants to rephrase the researcher's requests in their own words so that they can check to see if they understand what is being asked of them. This strategy aims to encourage participant autonomy and make sure they comprehend the terms of the research before giving their consent: "…*But is to say, "what we've asked of you here, could you translate that into your own words? And would you then agree with that?". So that the participants themselves are able to say, this is how I understand you, is that correct? And if this is indeed the case, I'm going say yes to that, or I'm going to say no to that.*" According to the researchers, it is necessary to give more time and importance to training researchers on the consent-taking process, including role plays that stimulate real-life scenarios with research participants to practice and refine their approach.

## Guideline development

### Problem statement

Insights from these results informed recommendations for optimizing the informed consent process, aiming to promote participant understanding, autonomy, and engagement in research endeavors. Based on the analysis of the results from the ideation session, two important factors emerged for creating more culturally sensitive and improved informed consent processes (depicted in Fig 2).

First, the imbalance in power dynamics between researchers and research participants needs to be reduced. Second, the process must ensure that participants are genuinely "informed," meaning they fully understand all aspects before deciding whether to join the study. The goal for the guideline to be developed was therefore to reduce these power imbalances and enhance participants' knowledge and agency, ensuring they have the necessary information to make an informed decision about participating in the study.

### Initial draft of the guideline

The guideline suggests a multifaceted approach to enhance informed consent practices. It begins with a diagnostic survey to evaluate current methods and interest in alternatives, encouraging collaborative adaptation with CABs. It also contains tools like the Values Identifier to aid in identifying values that may be overlooked in consent materials and approaches, while role play scenarios deepen understanding of informed consent. Reflective discussions using Common Problems Cards and analyzing participant feedback to inform further

development. In addition, creating a wish list based on survey responses guides the adaptation or co-design of consent approaches, aiming for culturally sensitive and participatory processes.

The guideline also suggests involving CABs from the outset, ensuring community interests are prioritized. This involves recruiting community members as outreach staff, evaluating benefits with CABs, and addressing community concerns. Secondly, aligning consent procedures with participant values is crucial, achieved through co-designing materials with CABs and offering flexible consenting options. Clarity regarding the ongoing nature of consent is emphasized, encouraging clear communication channels and contextualizing withdrawal rights. Researchers are advised to tailor their approach to community needs, focusing on cultural sensitivity, ethical rapport, and validation of participant understanding. Continuous

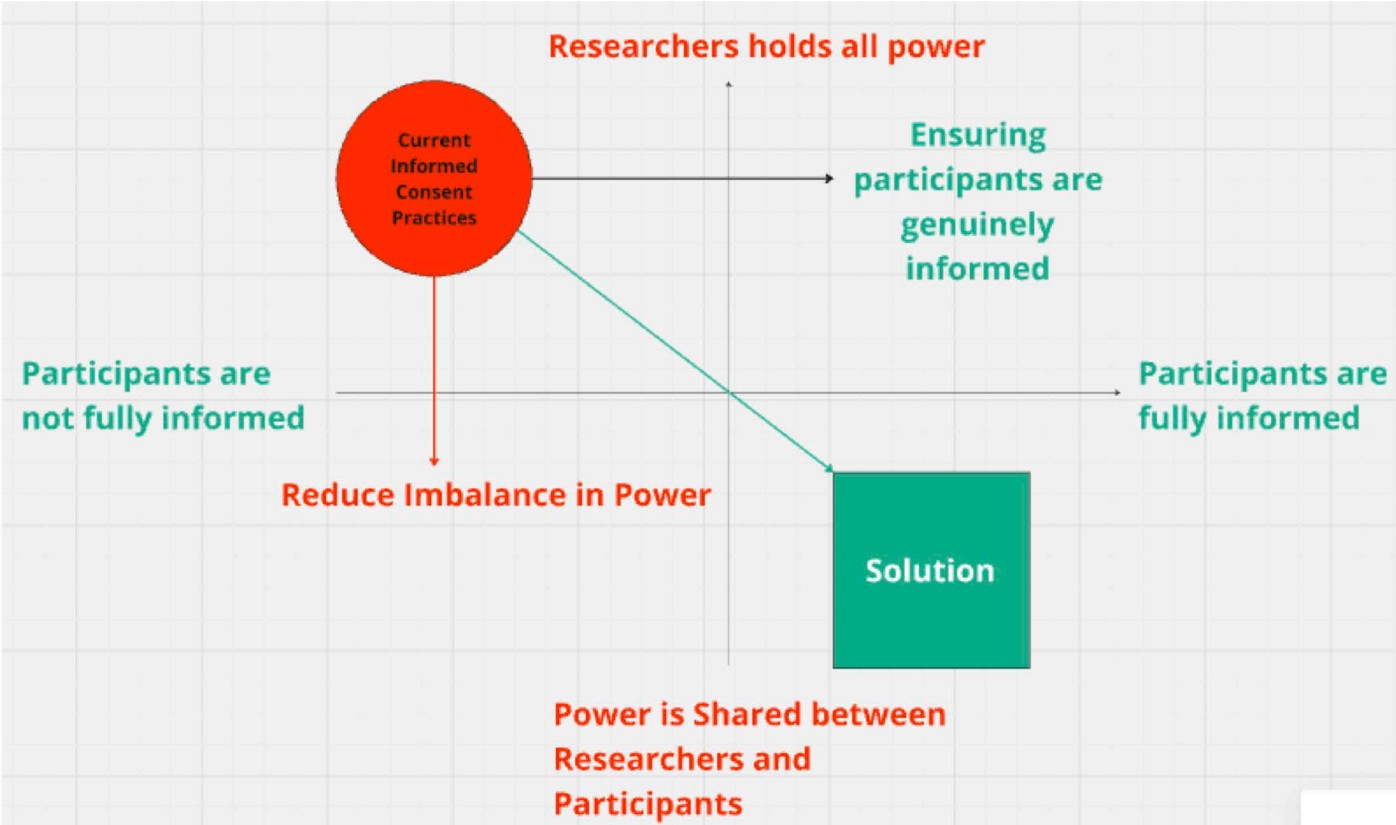

*This figure illustrates the contrast between traditional informed consent processes, where researchers hold most of the power and participants are not fully informed, and an improved approach. The proposed solution highlights the need to reduce the imbalance of power by fostering shared decision-making and ensuring that participants are genuinely and fully informed. This shift aims to promote ethical and equitable research practices.*

**Fig 2. Evolving informed consent: Towards equity and transparency.**

learning is promoted through seeking feedback, ethics audits, and fostering co-learning between stakeholders. Lastly, training in consent-related areas, including reflexive questioning and cultural competence, is recommended to ensure effective communication and understanding throughout the consent process. To support this work, we have provided a general check list for culturally relevant informed consent (S2 Checklist).

## Discussion

This study sought to bridge a notable gap in literature and practice by using a DT framework combined with PAR to collaboratively develop a culturally relevant informed consent guideline. By collecting insights from both researchers and affected communities, the study aimed to create a guideline to enhance informed consent for vulnerable populations in mental health research. Key themes identified include motivations for participation, the importance of trust, challenges in delivery and contextualization, perspectives on commitment and autonomy, the influence of gender and nationality, power dynamics in vulnerable populations, and recommendations for cross-cultural research.

Similar studies across different cultural and geographical contexts have highlighted both universal and context-specific challenges in informed consent processes. Research in Tibet emphasized the need for culturally sensitive procedures that respect local values [39], while a systematic review of cultural factors in health research stressed the importance of considering cultural nuances to protect participants' autonomy [40]. Additionally, studies on cross-cultural research methodologies point to the necessity of adapting consent procedures to specific cultural contexts [41]. The challenges of informed consent are highlighted in diverse settings, stressing the importance of clear communication and respect for local beliefs [42], while ethical dilemmas are seen as arising from power imbalances that influence participation decisions [43]. Furthermore, the need to adapt consent processes to participants' literacy levels and language preferences is underscored in low-resource settings [12]. These findings resonate with our study, which revealed that low literacy, language barriers, and power dynamics significantly influence research participation. Addressing these barriers through tailored communication strategies and supportive facilitation can help ensure truly voluntary consent.

In Lebanon, cultural and social factors significantly influence the informed consent process, especially among displaced populations. Language barriers, particularly among refugees speaking different dialects, can hinder understanding of research materials, making it crucial to adapt consent forms to local dialects and literacy levels [44]. Family dynamics also play a key role, as individuals may feel pressured to seek family approval before participating, potentially limiting autonomy [44]. Additionally, religious beliefs, common across Lebanon's diverse population, can affect willingness to engage in certain research, particularly if it conflicts with religious values. For displaced persons, perceived coercion may arise from fears of jeopardizing access to aid, further complicating the informed consent process. Researchers must be mindful of these cultural nuances to ensure voluntary and informed participation, particularly for vulnerable groups like refugees [44].

Before exploring the themes, it is important to define culturally relevant informed consent based on this study. We define it as an ongoing, dialogical process in which individuals from diverse cultural backgrounds are provided with information in a manner that is culturally sensitive, clear, and accessible. This process respects the participant's values, beliefs, and language while recognizing cultural differences in communication and decision-making practices. It emphasizes open, continuous communication, ensuring that participants are not only informed but also able to make voluntary and well-understood decisions that align with

their cultural context. Importantly, informed consent is not a one-time event but a dynamic interaction that evolves throughout the course of participation.

One of the first key findings that shape and influence this process of culturally relevant informed consent in practice is exploring the motivations behind consenting to participate in research. This is a critical yet often overlooked aspect in research literature. Understanding why individuals participate is key. Building a dialogue with participants is vital for grasping their perspectives. This strengthens researcher-participant relationships [22]. Our findings indicate that the use of dialogue in research is deemed ethically justified as it encourages respect, empathy, and genuine voices, making it a valuable approach to address sensitive societal issues. To delve deeper, comprehending the reasons behind a participant's affirmative response to a study was found to be essential. Equally significant was the exploration of the factors preventing participation, such as specific circumstances or personal reasons. Researchers must exercise caution to avoid manipulation, emphasizing the provision of necessary information and resources to enable participants to give genuine informed consent [10]. Flexibility and alternative choices were discovered as integral components of a participant-centric approach. We recommend that researchers should anticipate and adapt to unforeseen circumstances, offering alternative options if participants need to modify their level of involvement or withdraw due to other priorities. It would also be ideal if this proactive approach to risk management is implemented from the study's inception, ensuring effective handling of any unexpected challenge.

While power dynamics in research participation have been acknowledged, a more explicit discussion on coercion and undue influence is necessary. Refugees often live in precarious socio-economic conditions, relying on humanitarian aid and services provided by NGOs and governments [45]. This dependency can create an implicit pressure to participate in research, as individuals may fear that refusal could jeopardize access to essential resources or future support [46]. Even when researchers do not intend to coerce, the structural conditions surrounding refugees can lead to situations where participation is perceived as obligatory rather than voluntary [47].

Several factors contribute to this implicit coercion: some refugees may feel a sense of obligation [48], a fear of repercussions—whether real or perceived [49] and language barriers and cultural differences may hinder full comprehension of the voluntary nature of research, further exacerbating the risk of undue influence [47]. Our study found that language and literacy barriers, and associated power dynamics in IC actually present significant challenges. Trained interpreters and community members can help bridge this gap [50,51]. Additionally, we advise involving individuals from the community of participants as Research Assistants (RAs) or members of a Community Advisory Board (CAB) to serve as interpreters or liaisons. This strategy would bridge the language gap and foster improved understanding. In addition, researchers can implement strategies such as using visual aids (e.g., pictograms, infographics) [52], verbal explanations in participants' preferred languages [53] and simplified written materials. Interactive approaches, such as role-playing or demonstration videos, can also enhance comprehension.

Additionally, refugee populations are frequently studied, which can lead to research fatigue and feelings of exploitation [48]. In such cases, individuals may participate out of resignation rather than genuine willingness, believing they have little agency in the process. The cumulative impact of these factors highlights the ethical challenges in ensuring that informed consent is truly voluntary and free from coercion [46]. Addressing these concerns requires researchers to adopt culturally sensitive approaches, including transparent communication, active engagement with participants, and ensuring mechanisms for freely declining participation without consequence [49].

Trust-building was noted as central to research participation. Notably our findings showed that long-term studies require different trust dynamics than short-term ones. The focus is on the pivotal role of building and sustaining relationships with research participants, underscoring how the quality of these connections significantly influences the overall success of the research endeavor [54]. We suggest that the approach to cultivating trust and relationships may also vary depending on the nature of the research. A fundamental principle is to be stressed on: regardless of the research type, there is an imperative need for unwavering trust, professionalism and compassion towards participants [54] This involves rejecting the reduction of participants to mere numbers or statistics, recognizing their intrinsic value in contributing to the research process. We also recommend acknowledging and appreciating participant inputs as participants should not only feel seen but also recognized and valued for their invaluable contributions to the research, underscoring their significant role in the overall research process.

Another important key finding was addressing the timing of obtaining informed consent, delving into questions of how much time should be allocated for this process and identifying the most appropriate moment to approach participants for their consent. Moreover, the dilemma of participants extensively involved in previous research studies necessitates a thoughtful approach to handling such cases and examining the ethical considerations surrounding repeated participation and potential participant fatigue. There is a variation in perspectives and understanding of the informed consent process between researchers and participants such as the use of the Participant Information Leaflet/Informed Consent length and timing [6]. One solution could involve employing concise and straightforward consent forms. However, when the effects of using shorter and simpler consent forms on comprehension and satisfaction among research participants was investigated, the study confirmed that neither length nor complexity significantly affects comprehension or satisfaction, emphasizing the need for further research on consent form effectiveness [55].

The comparison between Informed Consent and job contracts is a novel finding. Some recommendations included clarifying the Informed Consent process and ensuring that participants have a comprehensive understanding, including their right to withdraw at any point. Enhancing communication was identified as another key recommendation, urging researchers to maintain an open and transparent dialogue with participants, addressing concerns, questions, and unexpected developments during the study. This proactive communication could foster a sense of informed decision-making and control among participants; however, more research should be done in this area. In addition, establishing an ongoing feedback mechanism was proposed as a solution, providing participants with a platform to express their concerns and expectations throughout the research process; the effectiveness of such an intervention should be investigated as well. Educating participants about their rights and the ethical principles governing research is found to be essential. We also recommend that the researchers should also maintain flexibility, acknowledging that participants' feelings and decisions may evolve over time. The goal is to help participants feel empowered, knowing they have control over their involvement and can withdraw without fear of repercussions. These recommendations collectively aim to bridge the existing gap in literature and contribute to the development of ethical and participant-centric practices in both informed consent and job contract contexts.

The examination of the role of gender, nationality, and community support in research participation underscores a gap in the existing literature. This gap raises concerns about the potential implications of not addressing it. For example, in our study, it was noted by both participants and researchers that engaging men in research activities can be challenging. This difficulty extends to ensuring their commitment to research participation, particularly in

studies involving vulnerable populations within the context of humanitarian work. One key recommendation is designing and conducting research that aligns with cultural norms, fostering inclusivity and respect for gender dynamics. Additionally, training research teams on cultural sensitivity and competence in handling gender-related issues is proposed.

Finally, within the realm of cross-cultural research additional recommendations were offered on how to improve the consenting processes, The adoption of audio-visual methods for consenting emerged as a notable strategy [56]. This approach recognizes the diverse linguistic and literacy challenges that vulnerable populations may face. Furthermore, the "Teach Back Method," as elucidated in the literature, provides an insightful avenue for enhancing understanding [57,58]. This method would involve participants explaining the information back to researchers in their own words, serving as a valuable tool to confirm comprehension and facilitate a more interactive and participant-centered informed consent process. These innovative strategies not only contribute to the discussion on power dynamics but also offer practical solutions for promoting inclusive and culturally sensitive research practices, particularly when working with vulnerable populations in cross-cultural contexts.

Participatory action research involving both researchers and potential research participants in co-creating a guideline proved to be both feasible and enjoyable for participants. This collaborative approach ensured that the guideline incorporated diverse perspectives, including those of the researchers and the participants, leading to a more comprehensive and relevant outcome. The process highlighted the value of inclusive research methods, as the guideline benefited from the combined insights and experiences of all stakeholders involved, ultimately enhancing its usability and effectiveness.

**Limitations.** The generalizability of the findings is limited due to using a single country setting of Lebanon and the cultural and contextual factors inherent in the country and research population. In addition, the measurement of participants comprehension and satisfaction with the informed consent processes mostly relied on self-reporting which potentially could introduce bias and inaccuracies. Moreover, the nature of this study may have influenced the participants' understanding and perception of the consent processes. Participants may have been more inclined to respond favorably due to the inclusive methodology, thereby affecting the objectivity of the findings. Finally, the prototype's effectiveness and feasibility in real-world contexts were not tested and this will be done in future studies.

## Conclusion

Informed consent emerges as an ongoing dialogue and partnership, encompassing effective communication, support, and continuous decision-making. The proposed solutions prioritize cultural sensitivity, autonomy, and transparency, fostering inclusive and respectful research engagements. These efforts underscore the importance of ethical reflection and continuous improvement, promoting meaningful interactions grounded in informed consent principles. To further advance the discourse, future research directions could explore the testing of the informed consent guideline and address other limitations encountered in this study. Such inquiries would extend the conversation and guide further exploration in the field. In conclusion, by embracing these solutions and fostering ongoing dialogue, research endeavors can aspire to more inclusive, respectful, and meaningful engagements that uphold the principles of informed consent, leaving a lasting impression on the ethical conduct of research.

## Supporting information

**S1 Checklist. COREQ checklist.**
(PDF)

**S2 Checklist. PLOS inclusivity questionnaire.**
(DOCX)

**S3 Checklist. Culturally relevant checklist.**
(DOCX)

## Acknowledgements

We are grateful for the time and advice from the Community Advisory Board members.

## Author contributions

**Conceptualization:** Emily Oliver, Imad Abou Khalil, Joseph Elias, Bassel Meksassi, Rozane El Masri, Thurayya Zreik, Bayard Roberts, Felicity Brown.

**Data curation:** Sandy Chaar, Emily Oliver, Rayane Ali, Imad Abou Khalil, Joseph Elias, Bassel Meksassi, Felicity Brown.

**Formal analysis:** Sandy Chaar, Emily Oliver, Joseph Elias, Bassel Meksassi, Thurayya Zreik, Felicity Brown.

**Funding acquisition:** Michèle Kosremelli Asmar, Bayard Roberts, Rabih El Chammay.

**Investigation:** Sandy Chaar, Emily Oliver, Rayane Ali, Imad Abou Khalil, Joseph Elias, Rozane El Masri.

**Methodology:** Emily Oliver, Rayane Ali, Felicity Brown.

**Project administration:** Emily Oliver.

**Supervision:** Emily Oliver.

**Writing – original draft:** Sandy Chaar, Emily Oliver, Thurayya Zreik, Felicity Brown.

**Writing – review & editing:** Emily Oliver, Rayane Ali, Imad Abou Khalil, Joseph Elias, Bassel Meksassi, Rozane El Masri, Thurayya Zreik, Michèle Kosremelli Asmar, Bayard Roberts, Rabih El Chammay, Felicity Brown.

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
